



# Benchmark dataset for hydraulic simulations of flash floods in the French Mediterranean region

Juliette Godet[1,4,*], Pierre Nicolle[1,*], Nabil Hocini[2], Eric Gaume[1], Philippe Davy[3], Frederic Pons[2], Pierre Javelle[4], Pierre-André Garambois[4], Dimitri Lague[3], and Olivier Payrastre[1]

[1]GERS-LEE, Univ Gustave Eiffel, IFSTTAR, 44344 Bouguenais, France
[2]Cerema Méditerranée, 13290 Aix-en-Provence, France
[3]Géosciences Rennes, Université Rennes 1, CNRS, UMR 6118, 35042 Rennes, France
[4]RECOVER, INRAE, Université d'Aix-Marseille, Aix En Provence, France
[*]These authors contributed equally to this work.

**Correspondence:** Pierre Nicolle (pierre.nicolle@univ-eiffel.fr)

**Abstract.** The absence of validation or comparison data for verifying flood mapping methods poses a significant challenge in developing operational hydraulic approaches. This article aims to address this gap by presenting a benchmark dataset for flash flood mapping in the French Mediterranean region. The dataset described in this paper (Nicolle et al., 2024) includes flood hazard maps and simulation results of three actual flash flood events, all computed in steady regime at a 5-meter resolution
using a 2D SWE model (neglecting inertia) named Floodos (Davy et al., 2017). Additionally, it includes the input data necessary (Digital Terrain Models, inflow discharges, hydrographic network) for conducting similar simulations with other hydrodynamic modeling approaches, in both steady and unsteady regimes. A comprehensive validation dataset, comprising observed flood extents, high water marks, and rating curves, is also provided, enabling a detailed evaluation of 2D hydraulic simulation results. The simulation results from Floodos, compared against stage-discharge rating curves available at gauging stations,
yielded highly encouraging outcomes. The median error (sim. - obs.) was -0.04 m for the 2-year return period and -0.14 m across all simulated return periods, ranging from 2 to 1000 years.

## 1 Introduction: context and objectives

Flash floods are a weather-related type of natural hazard, which, in Europe, most frequently occur in the Mediterranean region. Some sub-regions in the north-western Mediterranean seem to experience more frequent extreme discharges (Gaume et al.,
2016). Several of these "hotspots" have been identified in the French Mediterranean area, which has been hit by many disastrous flash floods in the last decade. Recent examples of these events include the 2019 floods in the Var, Alpes-Maritimes, and Vaucluse departments, which caused 13 fatalities and up to EUR 450 million in insured losses, or the 2020 "Storm Alex" event in the Alpes-Maritimes department, which caused 19 fatalities and up to EUR 230 million in insured losses (CCR, 2021). Although the effects of climate change on the intensity and frequency of flash floods in the French Mediterranean region remain
unclear (Tramblay et al., 2019), rapid urbanization and population growth in this area will likely increase the risk of disastrous flash floods. Therefore, the improvement of tools for flood risk management is in high demand, especially since many recent



studies emphasize the usefulness of information about potential impacts (Merz et al., 2020; Apel et al., 2022), which requires to develop enhanced flood mapping capacities.

Fortunately, the scientific context is conducive to tackling these issues, as flood mapping is being increasingly improved by recent developments in Digital Terrain Model (DTM) resolution and accuracy—especially thanks to Light Detection and Ranging (LIDAR) technology. For instance, LIDAR has led to a coverage of 99% of England at a 1m spatial resolution (Environment Agency, 2024), and the ongoing project LIDAR HD plans to provide DTMs, DEMs (Digital Elevation Models) and DSM (Digital Surface Models) for France at a 0.5m spatial resolution (IGN, 2024). Additionally, there have been recent advances in the estimation of river channel bathymetry in DTMs, which is unobserved in current products and represents a challenging issue for the future (Lague and Feldmann, 2020; Pricope and Bashit, 2023; Frizzle et al., 2024).

Current flood mapping methods typically rely on two main approaches. The first, commonly used, assumes that an area is flooded if the elevation of the floodplain cell is lower than an assumed or estimated flood water height in the channel, a GIS-based method known as DTM filling. For example, the Height Above Nearest Drainage (HAND) method (Nobre et al., 2011) estimates flood water height based on topographic relationships derived from the terrain and channel structure. This method offers a simple and computationally efficient way to approximate inundation extents. The second approach, which is the focus of this paper, is based on the numerical solution of two-dimensional shallow water equations (SWE), commonly referred to as hydraulic models. While SWE are typically solved in two dimensions, one-dimensional (1D) models have also been used, such as in Le Bihan et al. (2017); Lamichhane and Sharma (2018), due to their shorter computation times and reasonable accuracy. In fact, 1D models share some features with GIS-based methods: while they use a hydraulic model to estimate flood stage along the channel network, they similarly rely on a GIS operation to project water surface elevations onto the terrain to derive inundation extents. However, 1D models have limitations in representing complex floodplain flows. To address this, three-dimensional (3D) models are being explored (Luo et al., 2018), but their real-world application remains limited by complexity and the current lack of calibration and validation data (Bates, 2022). In short, hydraulic 2D SWE models offer a good compromise between data availability, accuracy, and computation times.

Continental and global-scale 2D flood mapping simulations have been largely performed since the early 2010s, most often at $10^2$ m resolutions. Examples include the 100m-European flood maps calculated by Alfieri et al. (2014) and updated by Dottori et al. (2022), and the Global Flood Maps calculated by Fathom, initially at a spatial scale of approximately 90m (Sampson et al., 2015; Andreadis et al., 2022). However, it can be argued that such large-scale models are not always relevant at local scales (Fleischmann et al., 2019), which are targeted for the representation of small rivers. Consequently, efforts are being made to develop methods applicable at finer resolutions ($10^1$ m). The latest version of Fathom's global maps (Wing et al., 2023) has been refined to achieve a spatial resolution of approximately 30m, with performance evaluated to be close to local model skill in many cases. Spatial resolutions lower than 10m are especially targeted for areas prone to flash floods, such as the 5m resolution used in Hocini et al. (2021).

Whatever flood mapping method is used, the assessment of its performance generally suffers from a lack of validation data, especially for the simulation of flash floods, which often occur in data-scarce regions. Crucial parameters that need to be assessed in flood mapping studies include flood extents, flood depths, and water velocities, all of which are rarely recorded



after a flash flood event for several reasons, including measurement difficulties (Molinari et al., 2019). Validation methods based on reproducing actual flood events may use observed extensions of the flooded area (Dottori et al., 2017; Hocini et al., 2021; Wing et al., 2023), high water marks (Hocini et al., 2021; Wing et al., 2023), as encouraged by Bösmeier et al. (2022), or
flood impacts (Le Bihan et al., 2017; Ritter et al., 2021), and sometimes gauge measurements (Gebrehiwot and Hashemi-Beni, 2022). Another possible data source for the validation of hydraulic models are the results of simulations performed by experts (Alfieri et al., 2014; Sampson et al., 2015; Albano et al., 2020; Hocini, 2022; Dottori et al., 2022; Wing et al., 2023). It can consist of historical event simulations or flood hazard maps (e.g., those produced as part of the European Flood Directive). In both cases, the main limitation lies in the input data, which sometimes are unavailable or undocumented. Without access to the
original input data used in a study, it becomes challenging to conduct a relevant comparison between models.

The aim of this article is to help fill the gap in validation data by providing a benchmark dataset for flash flood mapping in the French Mediterranean region. Figure 1 specifies the content of the dataset and the corresponding structure of this paper, which is separated in two parts: flood hazard and flood events. On one hand, we provide inputs for flood hazard calculations in steady state, with an example of results obtained with the Floodos model (section 2.2.1), as well as a large sample of rating
curves usable for validating the flood hazard maps (section 2.2.2). On the other hand, we provide inputs for simulating three historical flash flood events, as well as the corresponding Floodos simulation results (section 2.3.1), and the observed flood extents and/or high water levels used for the validation (2.3.2). The provided data can be used to run and evaluate similar simulations (flood hazard maps or specific flood events) with any other hydrodynamic modeling tool. In this paper, we also present the production methods, including the DTM preprocessing method (section 3.1), the Cinecar rainfall-runoff model
(section 3.2) and the Floodos method (2D SWE-based approach, without inertia, detailed in section 3.3). Finally, we illustrate the validation results obtained with the Floodos approach (section 4).

## 2   Data description

This section aims to describe the provided data (see table A1 for an exhaustive list of the data made available). We first present the areas of the French Mediterranean region covered by the simulations. Then, following the structure of the dataset, we
describe successively the data related to flood hazard maps and the data related to historical flood events. In each of these two sections, the data are presented according to their role in hydraulic modelling: input/output or validation data.

### 2.1   Study areas

The whole simulation area covers 61 elementary hydrological areas (watersheds grouping together nearly 300 sub-basins of sizes between 20km$^2$ and 750km$^2$, 140 km$^2$ on average) located in the French Mediterranean region, which represent nearly
42,000 km$^2$ of drained area and 20,000 km of river network (see figure 2). The Rhône and Durance rivers were excluded from the study area because they are seldom impacted by flash floods and are significantly influenced by artificial structures, such as dams. The area covers a wide range of climatic patterns, hydrological and geological properties, and population densities. It has been subject to numerous flash floods in the last decades, especially the floods in the Argens watershed (15 June 2010),





**Figure 1.** Dataset structure and organisation, and related sections of the data paper. The signification of each item is summarised in table A1.

in the Alpes Maritimes department (3 October 2015), and in the Aude watershed (15 October 2018), which are studied in this

paper. These floods led to up to EUR 710 million of insured losses and 25 victims for the most devastating event (June 2010 flood). Detailed descriptions of each event with meteorological and hydrological information can be found in Hocini et al. (2021), whose case studies, input and validation data have been integrated in this article.

## 2.2 Computation and validation of flood hazard maps

Flood hazard maps corresponding to eight return periods between 2 and 1000 years were generated on the whole simulation

area. The 2D Floodos computation results included in the dataset were obtained based on the following input data: the Digital

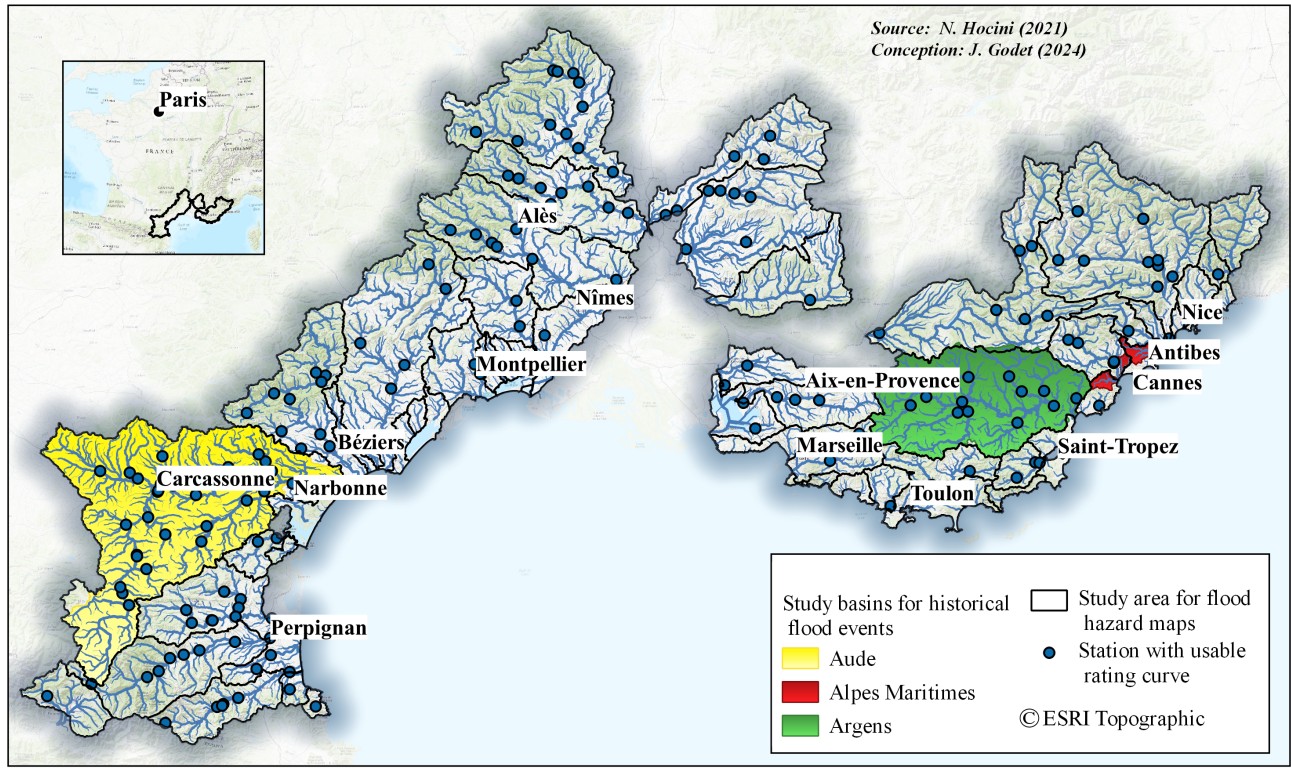

**Figure 2.** Study basins for the simulation of historical events (coloured), study areas for flood hazard maps (framed in black) and the selected stations for rating curve data.

Terrain Models (DTMs), the vector description of the hydrographic network, and the peak discharges for each return period. For the asessment of modeling results, we provide in the dataset rating curves from stream gauges in the study area. Figure 3 illustrates the computation and validation procedures of flood hazard maps.

### 2.2.1 Flood hazard mapping inputs and outputs

The DTM used is the RGE ALTI® produced by IGN (Institut national de l'information géographique et forestière, https:// geoservices.ign.fr/rgealti), with a spatial resolution of 1m, that includes the latest lidar measurements available in October 2022 (date of extraction from IGN repository). It must be noted that the entire study area was not covered by a lidar measurement at the time of the calculations (see Figure 10). This DTM provides average altitude information for each 1m × 1m pixel. It was used as is for the DTM cleaning process (see section 3.1), and it was resampled at the resolution of 5m × 5m for the hydraulic

calculations. The 5m DTM directly obtained from the resampling is referred to as "raw", while the 5m DTM that underwent a cleaning process is referred to as "processed".





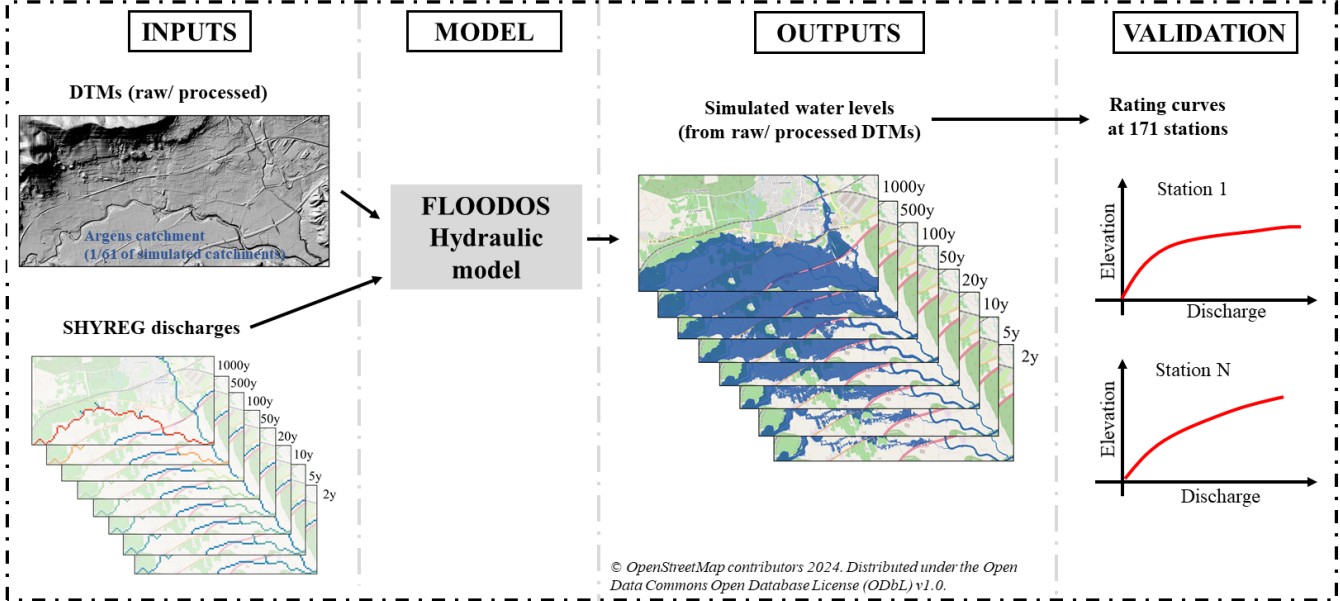

**Figure 3.** Illustration of the computation and validation of flood hazard maps

The hydrographic network provided in the dataset is a raster description of the streambeds, derived from BD TOPAGE®, which is the French reference hydrographic network database. Based on the provided raster description of the streambeds, two different Manning's friction coefficient values were used in the hydraulic simulations: $n = \frac{1}{10} s.m^{-1/3}$ for the flood plains, and

$n = \frac{1}{18} s.m^{-1/3}$ for the streambeds, for all river reaches. These values have been determined with a sensitivity analysis led by Hocini (2022) in the study area.

In order to carry out modelling of flood hazard over the whole computational domain, it is necessary to gather discharge data for defining the inflow boundary conditions of the hydraulic model. This discharge data was extracted from the SHYREG (REGionalized SHYpre model) database, and is included in the dataset. SHYREG is a national database providing synthetic

rainfall intensity and peak discharge quantile estimates for the whole French territory (Arnaud et al., 2014; Aubert et al., 2014). The SHYREG quantiles are derived from stochastic simulations combining an hourly stochastic rainfall generator and a simplified distributed rainfall–runoff model, respectively calibrated against the existing raingauge and streamgauge measurements. The database includes the peak discharge quantiles for 8 return periods of 2, 5, 10, 20, 50, 100, 500, and 1000 years. These discharge quantiles are provided continuously along the hydrographic network, for drainage areas larger than 5 km$^2$, on a 50

m resolution grid. Therefore, this method can provide discharge information for small ungauged river reaches, which is crucial for evaluating risks related to flash floods. Like all methods, estimations provided by SHYREG are subject to uncertainty, due to major simplifications in the model, concerning, for instance, the influence of hydraulic regulation structures, karstic zones, rain and discharge data uncertainty, etc. Caruso et al. (2013) thus provide confidence levels to qualify the applicability



of the method on each stream network in France. Details on how the SHYREG quantiles were incorporated into the Floodos computational domains can be found in section 3.3 and appendix D.

The flood mapping procedure described in section 3.3, applied to the aforementioned data, yielded $61 \times 8$ water level rasters (one for each of the 61 sub-domains of figure 2 and for each of the 8 return periods) at the 5m $\times$ 5m resolution. Initial results, derived from the same hydraulic modelling approach but with different input data (i.e. different DTMs that were pre-processed using a much more simplified approach than the one described in section 3.1), showed a large overestimation of the flooded

areas for the 2-year return period, mainly attributed to residual defects in representing the geometry of the streambeds in the DTMs (Hocini, 2022). This justifies the application of a specific DTM processing to improve the representation of the streambeds. Figure C1 illustrates the impact of this processing procedure on river streambeds. For the sake of comparison and illustration of the efficiency of the DTM preprocessing method, results obtained from both raw and processed DTM are included in the dataset. A quantitative assessment of the results based on stage-discharge rating curves available at gauging

stations over the studied river networks is carried out in section 4.1, and shows that the DTM preprocessing method largely address the overestimation for low return periods.

### 2.2.2  Validation data : Rating curves

As discussed in the introduction, the assessment of a set of flood hazard maps for different return periods is highly limited by the lack of validation data. The proposed approach leverages the expertise of local hydrometry services, which is summarized in

the rating curves available at streamgauge river sections. Such an evaluation was already applied by Le Bihan et al. (2017), who examined gauging data and rating curves from twelve hydrometric stations. It was applied here for a much broader assessment, at a regional scale; on 171 gauging stations selected as follows. Initially, both gauged (i.e. measured) discharges and rating curves were considered. However, due to the very limited number of measured discharges corresponding to return period exceeding 2 year (only 2% of the total amount of gauged discharges, spread over 35% of the gauging stations), it was decided

to focus on the rating curves. The rating curves are also subject to uncertainty, as they are derived from measured discharge data. However, we argue that they serve as a convenient and independent source of data for evaluation. They were provided by SPC Grand Delta, SPC Méditerranée Ouest, and SPC Méditerranée Est, either through the open-access Hydroportail platform (https://hydro.eaufrance.fr/, last accessed: April 7th, 2024), or directly by the data producers through the BAREME tool (Bechon et al., 2013), when the data was not available on the Hydroportail platform (representing roughly 1/3 of the stations). The

data provided in the dataset do not correspond to the raw data extracted from Hydroportail/BAREME - they have undergone a procedure to provide uniform (data extracted from Hydroportail are not formatted the same way as data extracted from BAREME), verified (some errors in the data could be detected), suitable (for flood hazard maps verification purposes), and simplified (keeping only the variable of interest) information. The procedure is detailed in appendix B.

The application of this procedure reduced the initial number of gauging stations of interest from 418 to 171. The 171 selected

stations are mapped in Figure 2, and Table 1 summarises the number of stations for which the rating curve providing estimated stage-discharge values for each of the eight estimated return periods.



| T (years) | 2 | 5 | 10 | 20 | 50 | 100 | 500 | 1000 |
|---|---|---|---|---|---|---|---|---|
| N | | 171 | 164 | 155 | 140 | 112 | 85 | 33 | 20 |

**Table 1.** Number of stations for which the rating curve has stage-discharge values exceeding each return period.

The data provided in the dataset include metadata such as station name, code, location, reference altitude of the water level gauge, etc., but also a "Quality index for high flow data" which is a qualitative score (between 1 and 3, 3: reliable data, 2: moderately reliable data, 1: uncertain data) that was estimated based on discussions with the local forecasting services. It
quantifies the precautions that need to be taken when interpreting the gap between the model simulations and the rating curves for high return periods. Furthermore, reliability zones (i.e. water levels between which the data is deemed reliable by the local forecasting service) which were directly extracted from Hydroportail / BAREME are also provided. Finally, despite this careful data selection, multiple errors can remain and uncertainties cannot be avoided, and this is particularly true when working with data as sensitive as the rating curves, which are often extrapolated from direct gauge measurements below the 2-year return
period.

The provided rating curves can take two forms: either as "piecewise linear" curves based on linear interpolation of stage-discharge pairs, or as a "power function" $Q = \alpha \times (H - \beta)^{\gamma}$, where $Q$ is the discharge in $m^3/s$, $H$ the water stage over the zero reference altitude of the water level gauge, and $\alpha, \beta, \gamma$ are parameters to be determined. Rating curves are constructed across a large range of discharge rates, although gaugings may not be available for the entire range. Return periods are not provided by
local hydrometric services and are determined independently using the SHYREG method.

### 2.3 Computation and validation of actual extreme flood events simulations

Hocini et al. (2021) compared three automated methods for flash flood inundation mapping, based on the simulation results of three observed flood events. In this section, we present the data associated with these three events (see Figure 4 for an overview of each dataset's role in the simulation chain). The input discharges and validation data (observed inundation extents and high
water marks) are shared with the study by Hocini et al. (2021). However, the input DTMs and the resulting simulated water levels differ, highlighting the impact of DTM preprocessing on hydraulic results

### 2.3.1 Flood mapping inputs/ outputs

DTMs and hydrographic networks are the same as those described in section 2.2.1, except that the Siagne and Brague watersheds were merged to enable the simulation of the Alpes Maritimes event. Manning's friction coefficients were set to
$n = \frac{1}{18} s.m^{-1/3}$ in the streambed and $n = \frac{1}{10} s.m^{-1/3}$ in the flood plains. Computations were performed in steady-state regime, using simulated peak discharges from Cinecar rainfall-runoff model (see section 3.2 for more details). Discharges are considered homogeneous on each river reach, of average length lower than 2km. We provide the full hydrographs at each river reach outlet, as well as the lateral inflow hydrographs, to enable unsteady computations for future users.





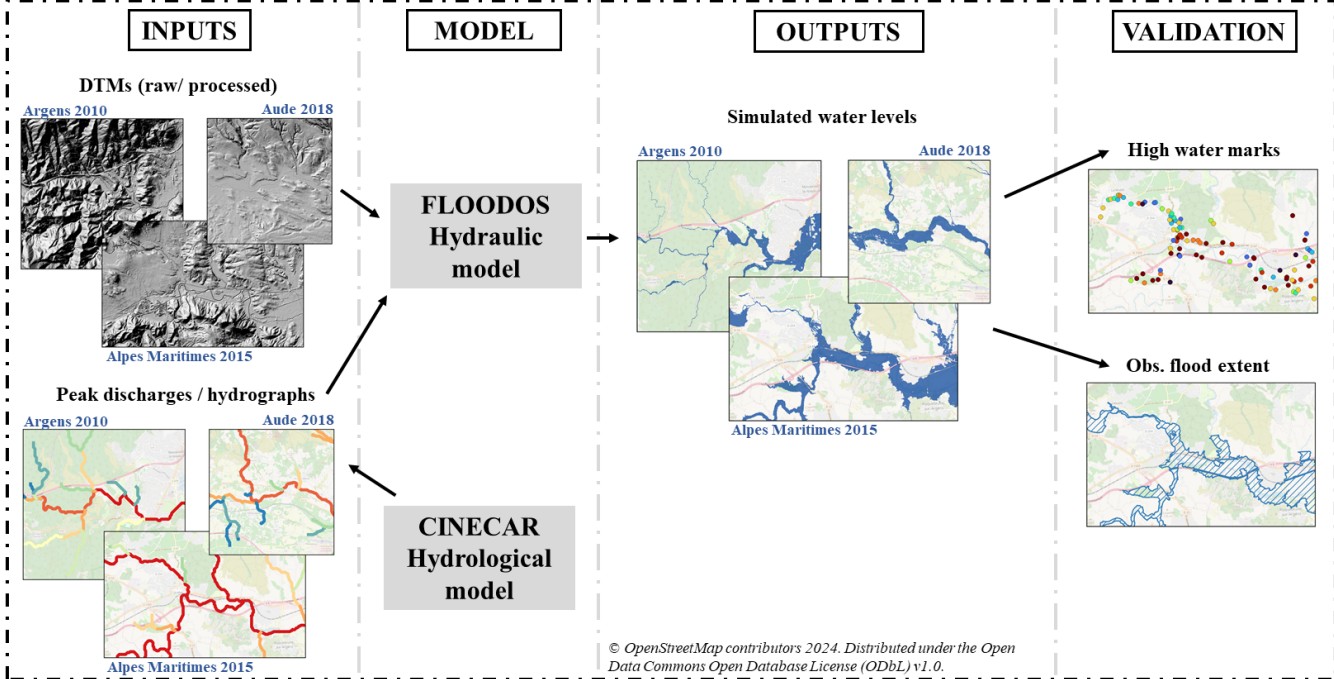

**Figure 4.** Illustration of the simulation and validation of extreme flood events

The water level maps (flood inundation extents and depths) generated with the Floodos model that are included in the dataset,
correspond to the maximum of inundation according to the peak discharge of the flood, they may thus correspond to different
instants of the flood for the various reaches of the river network.

### 2.3.2   Validation data : Flood marks and flood extents

As mentioned earlier, the validation data used to assess the quality of flood event simulations originate from the study by
Hocini et al. (2021). However, they are properly described and published for the first time in this datapaper. We also repurpose
these data to compare the impact of DTM preprocessing on hydraulic results (see Section 4.2). Two types of data are provided:
high water marks (HWMs) and reconstituted flood extents from field observations.

HWMs were extracted from the French HWM database (https://www.reperesdecrues.developpement-durable.gouv.fr/, last
access: April 19th, 2024). We provide 642 HWMs for the 2010 Argens event, 556 HWMs for the 2015 Alpes Maritimes event,
and 1089 HWMs for the 2018 event. To ensure that collected HWMs only correspond to river flooding, HWMs were manually
filtered according to their proximity to the modelled river network. While these data should not include large errors as they
were systematically checked, they are not immune to occasional problems. For instance, amongst the HWMs collected for the
2010 Argens flood event, systematic errors in altimetric referencing were observed on a 28km-long river reach of the Argens
river, running upstream from the north-west of the town of Vidauban. It was particularly obvious since some HWMs were





clearly under the natural terrain level, as revealed by a simple comparison with the DTM. All these detected errors concerned
HWMs surveyed by the same firm at the same date. While these errors may have been caused by human error, other ones
can also be the result of local obstacles affecting water surface elevation, capillary rise of moisture in walls, instrumentation
limitations, etc. The detected faulty HWMs were systematically removed from the dataset.

The reconstituted flood extents provided in the dataset are also a combination of maps obtained from several firms of
engineering consultants, commissioned by local authorities. Reconstitution methods were generally based on field surveys,
inventories and locations of HWMs, photos, videos, and satellite images. Limits can come from local interpolations between
field observation points. These maps are available for the Argens 2010 event and the Aude 2018 event, however the Alpes
Maritimes 2015 event was characterised by very rapid kinetics in urban environments, thus it was more complicated to conduct
the same reconstitution procedure. Local authorities provided a map which are partly the result of hydraulic simulations, thus
it was decided not to use it for validation purposes. The flood extent dataset also provides a shapefile of subcatchments under
study, which were limited to the river network where the observed flooded areas are available. These subcatchments were
calculated based on a 50m flow direction grid and have an average area of $2.5km^2$. They were designed in order to conduct the
evaluation process at a local river reach scale.

## 3 Description of data prepocessing methods

### 3.1 DTM preprocessing

As mentioned in section 2.2.1, a specific processing of the DTMs was applied to attenuate or remove several imperfections
suceptible to cause large errors in hydraulic computations: imperfect interpolations of DTMs in stream beds leading to noisy
cross-sectional and longitudinal bathymetric profiles, missing description of the capacity of covered watercourses and pres-
ence of undesirable obstacles to flow in river beds (typically bridges decks incompletely removed). These imperfections may
generally lead to overestimate flooded areas particularly for the low (i.e. 2y) return periods.

The DTM processing procedure was based on the work of Kalsron et al. (2019) to "reopen" of covered river reaches, and on
the methods of Wimmer et al. (2021) for the delineation of the streambed. They are thoroughly described in a technical report
to the French Ministry for the Environment (Nicolle et al., 2023), and shortly presented in appendix C.

The effects of these DTM corrections on hydraulic computation results were analysed in three main steps. First, hydraulic
modeling results samples were visually inspected. An example is provided in figure 5 for the Brague river. It illustrates several
of the targeted imperfections. A better definition of the streambeds' cross-sections clearly limits simulated overflows for the
T=2 year floods (two comparisons on the top of the figure) or eliminates the overflow upstream a bridge not eliminated in the
original DTM (bottom of the figure). Then, the total number of flooded pixels outside the streambed for the 2-year maps were
counted: it should be as limited as possible since it is generally considered that the streambed has the capacity to convey the
2-year flood without overflow. On average on all the stream reaches, the number of inundated pixels outside the streambed
for the 2-year return period was reduced by 40% due the DTM processing (minimum decrease of 4%, maximum decrease of
95%, no increase). Finally, validation results for both configurations (raw DTMs and processed DTMs) were compared using





the validation data described in Section 2.2.2 (the rating curves). These results demonstrate the statistic relevance of the DTM correction method in river channels, as discussed in Section 4.1.

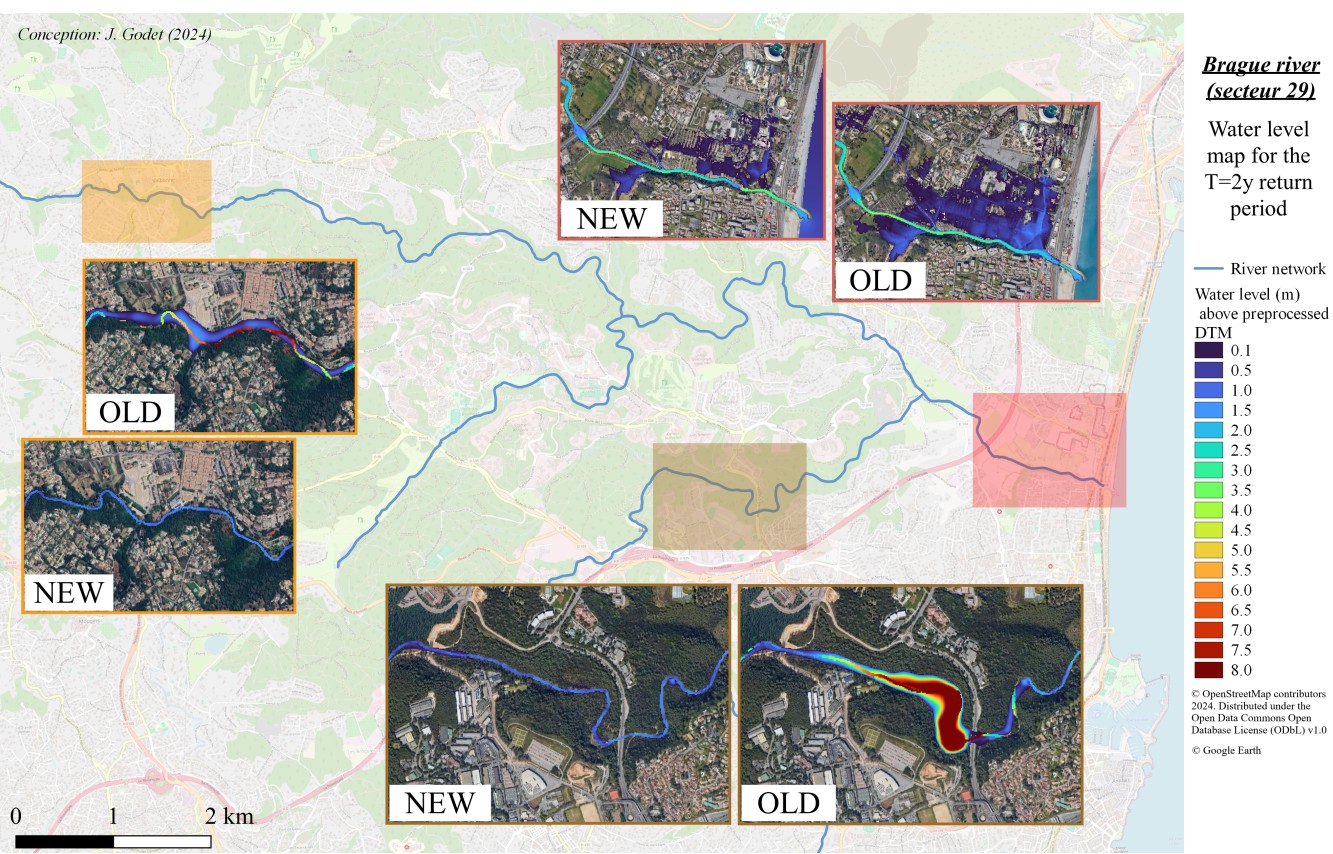

**Figure 5.** Comparison of simulation results obtained from raw and processed DTMs for the Brague river and the T=2y return period.

## 3.2 Hydrograph calculations

235 The dataset provides flood hydrographs for the three historical flash flood events under study (Argens 2010, Alpes Maritimes 2015, Aude 2018), for each of the considered river reaches. These were calculated using the Cinecar distributed rainfall-runoff model (Gaume et al., 2004; Naulin et al., 2013), calibrated for each flood event based on the available measured and estimated discharges. Cinecar is based on a division of the river network into river reaches (1.5km-long on average in the present cases) that are each connected to two rectangular slopes, representing the left and right bank subwatersheds. Cinecar simulates the 240 evolution over time of runoff coefficients and volumes on each subwatershed based on the Soil Conservation Service-curve number model (SCS-CN). The computation time step can be adjusted. It has been set to 15 min in this work. To route the effective rainfall through the watershed, the kinematic wave model is used on the hillslopes and in the river network, except for river reaches with slopes lower than 0.6%, for which the Hayami solution for diffusive wave model is used.





For the Argens 2010 and the Alpes Martimes 2015 events, Cinecar was forced with the Antilope J+1 rainfall product (Champeaux et al., 2009), which combines radar and rainfall records. It is an operational product provided by Météo-France the day following the precipitation record date, permitting an incorporation of 40% of additional data from rainfall network compared to the online Antilope rainfall product. This may not always prevent from observing important errors locally (especially in regions scarce in rain gauge networks), which is why the model was forced with an improved rainfall reanalysis for the Aude 2018 event (Caumont et al., 2021). This product takes advantage of new rainfall data, in particular amateur measurements, and is considered to be more reliable than the initial Antilope J+1 product. The original time step is 1h, thus the rainfall rates were uniformly distributed over the 15min sub time step to feed the Cinecar model. Cinecar was calibrated for each event using all available observations of peak discharges, including a large number of estimations at ungauged sites, gathered within the HyMeX program (Ducrocq et al., 2019). As Hocini et al. (2021) reported, the resulting relative difference between simulated and observed peak discharges rarely exceeds 20% (see figure 6).

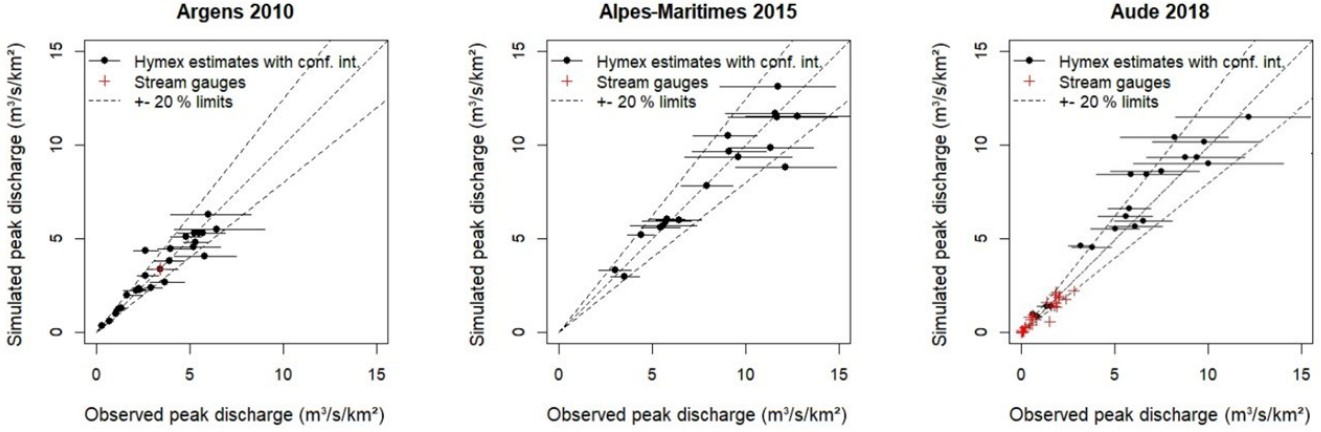

**Figure 6.** Observed vs. simulated peak discharges with the Cinecar rainfall–runoff model for the three simulated flood events (adapted from (Hocini et al., 2021)).

## 3.3 Floodos computations

Floodos is a 2D lagrangian hydrodynamic model solving the 2D SWE equations, neglecting inertia. It routes elementary water volumes (i.e. precipitons) over the topography, the number of precipitons entering the model per computation time step depending on the discharge (Davy et al., 2017). It is coded in C++ and it is distributed in the form of a Windows executable that can be launched from three input files representing the topography (GSBG file, ".alt"), the discharges to be injected (GSBG file, ".rain") and the parameters to be applied (text file, ".dat"). In this work, the routine including the generation of these files and launching the calculation has been wrapped in R.

Hocini et al. (2021) showed that for steady state simulations, Floodos led to more accurate results for 2D modeling of flooding than a 1D hydrodynamic approach (caRtino 1D) and than a DTM filling approach (HAND/MS). Floodos also has the





advantage of being relatively simple to implement, as the calculations are performed directly on the DTM mesh, and of being
very fast. However, inertia terms are neglected in the resolution, which can cause some errors in areas with abrupt changes in
direction and/or in the vicinity of obstacles in the flow. Additionally, the model requires careful verification of convergence,
which has been automated here. In this work, simulations are conducted independently for each river reach and are performed
in a steady-state regime based on flood peak discharges (inflow discharges, the hydrological/hydraulic coupling being described
in appendix D) provided by the SHYREG database (for the flood hazard maps) or by the Cinecar hydrological model (for the
historical flash flood events). The steady-state assumption can overestimate inundation extents and depths if the flood wave
volume is relatively small compared to the floodplain's storage capacity. However, this assumption is considered reasonable
here, as the floodplains are only a few hundred meters wide, and their storage capacities are therefore limited. Additionally,
computations based on peak discharges may overestimate backwater effects at confluences due to the assumption that peak
discharges occur simultaneously across all river branches at a confluence.

Details on the implementation of the Floodos numerical model over the whole French Mediterranean region is presented in
appendix D, and is derived from the previous works of Hocini (2022); Nicolle et al. (2021).

## 4    Illustration of the validation methods and of the results obtained with the Floodos model

### 4.1    Validation of flood hazard maps using hydrometric data (rating curves)

This section illustrates the use of rating curves for the verification of the flood hazard maps. In order to compare the rating curve
values to the simulated water levels, the first step consists in extracting from hydraulic computation data the simulated water
levels and discharge values at each gauging station location. To get the discharge values, the gauging stations were connected
to the nearest SHYREG pixel. Connection based on a distance criterion (150m here) might not be the most appropriate method
to determine the corresponding hydrological pixel of a gauging station - Godet et al. (2024) have documented this issue - but as
the gauging stations only concern intermediate to large catchments, it was considered that the risk of error was limited, though
all allocated positions were manually checked. The allocation gauging station / SHYREG pixel makes it possible to extract, for
each gauging station, discharge values $Q_{shy}(T)$ corresponding to each return period $T$. Then, we retrieve the simulated water
altitude $Z(x, y; T) = h(x, y; T) + b(x, y)$ with $h$ the water depth and $b$ the bathymetry elevation, at the original location of the
gauging station, i.e. not at the location of the allocated SHYREG pixel. $DEM_{alti}$ is the altitude of the pixel according to the
DTM. This process results in eight triplets $\{T, W_{alti}(T), Q_{shy}(T)\}$ (one for each simulated return period) for each gauging
station, which can be directly compared to the rating curve on a graph (see figure 7).

To complete the evaluation of the effect of the DTM preprocessing (see section 3.1) on hydraulic modelling results, simu-
lation results obtained from both raw and processed DTMs can be compared. Examples of such comparisons are provided in
figure 7. As illustrated in these examples, for a significant number of gauging stations there is good or very good agreement
between the theoretical rating curves established by the hydrometric services and the simulated water levels, with differences
often lower than 0.5m, even though largest errors can also be observed (e.g. the example of the station of Le Gapeau à Hyères).
The results do not indicate a systematic overestimation or underestimation the water levels, even if underestimation increas-

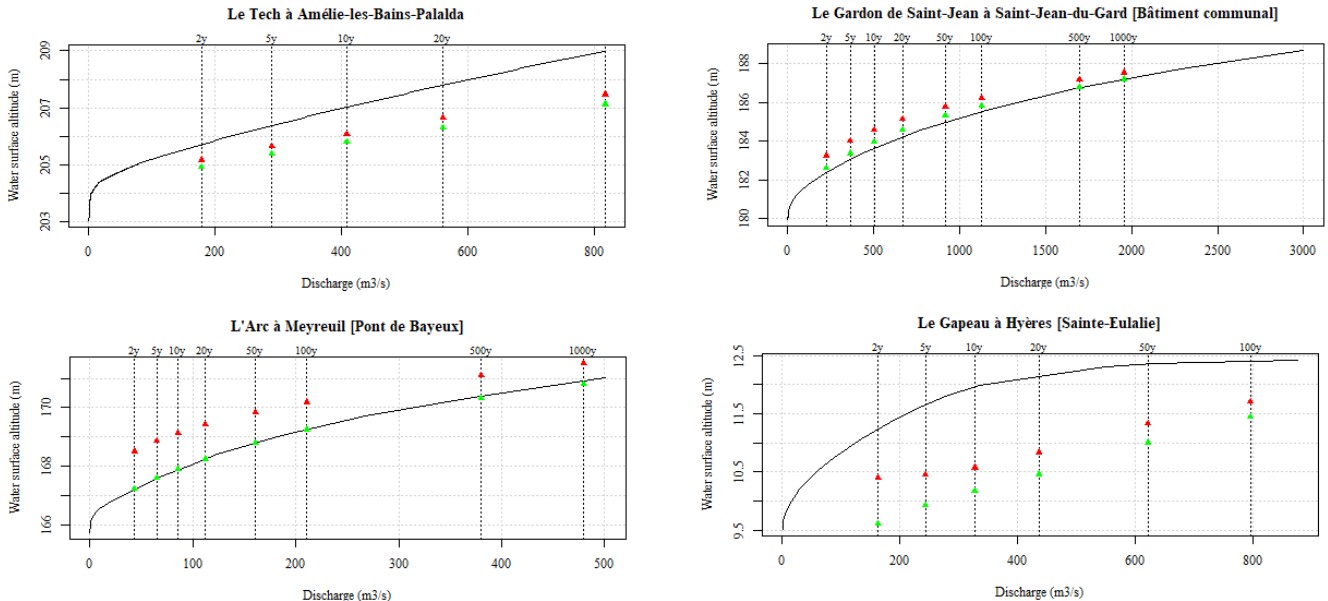

**Figure 7.** Examples of comparisons of theoretical rating curves elaborated by local hydrometric services (black line) and stage-discharge relations based on the regionally implemented Floodos model based on the raw (red points) or processed (green points) DTMs.

ing with the return period, like in the illustrated example of La Tech at Amélie-les-Bains-Palada, is observed several times. The example of the Arc à Meyreuil or the Gardon at Saint Jean illustrate the expected improvement induced by the DTM prepro-cessing. however such an improvement is not systematically observed, as illustrated in 2 out of 4 of the presented examples:
globally, when the water levels were already underestimated, the DTM pre-processing tend to exacerbate the inaccuracies in the results.

To provide a more comprehensive evaluation, boxplots of differences in water altitude are presented for each return period in Figure 8. This confirms that the general effect of the DTM preprocessing is a decrease in the simulated water altitudes. This was expected since the DTM preprocessing procedure described in section 3.1, leads to increase the hydraulic capacity of the
streambeds. This impact is noticeable for all return periods and leads to an overall improvement of the agreement between the expert-based rating curves and the simulated water levels for low to medium return periods: the median error for the T=2years is reduced from 0.54m (raw DTM) to -0.04m (processed DTM). The median error (sim. - obs.) for all return periods combined is of 0.38m for the water levels simulated using raw DTMs, and of -0.14m for the water levels simulated using processed DTMs. These results are encouraging given that they are of the same order of magnitude than typical errors observed for the
validation of flood events mapping using high water marks (see Hocini et al. (2021) and section 4.2), despite the fact that this verification method is very challenging. It is important to recall that the simulations are based on a numerical 2D model, implemented at a large regional scale with fixed friction parameters values and no further local adjustment.

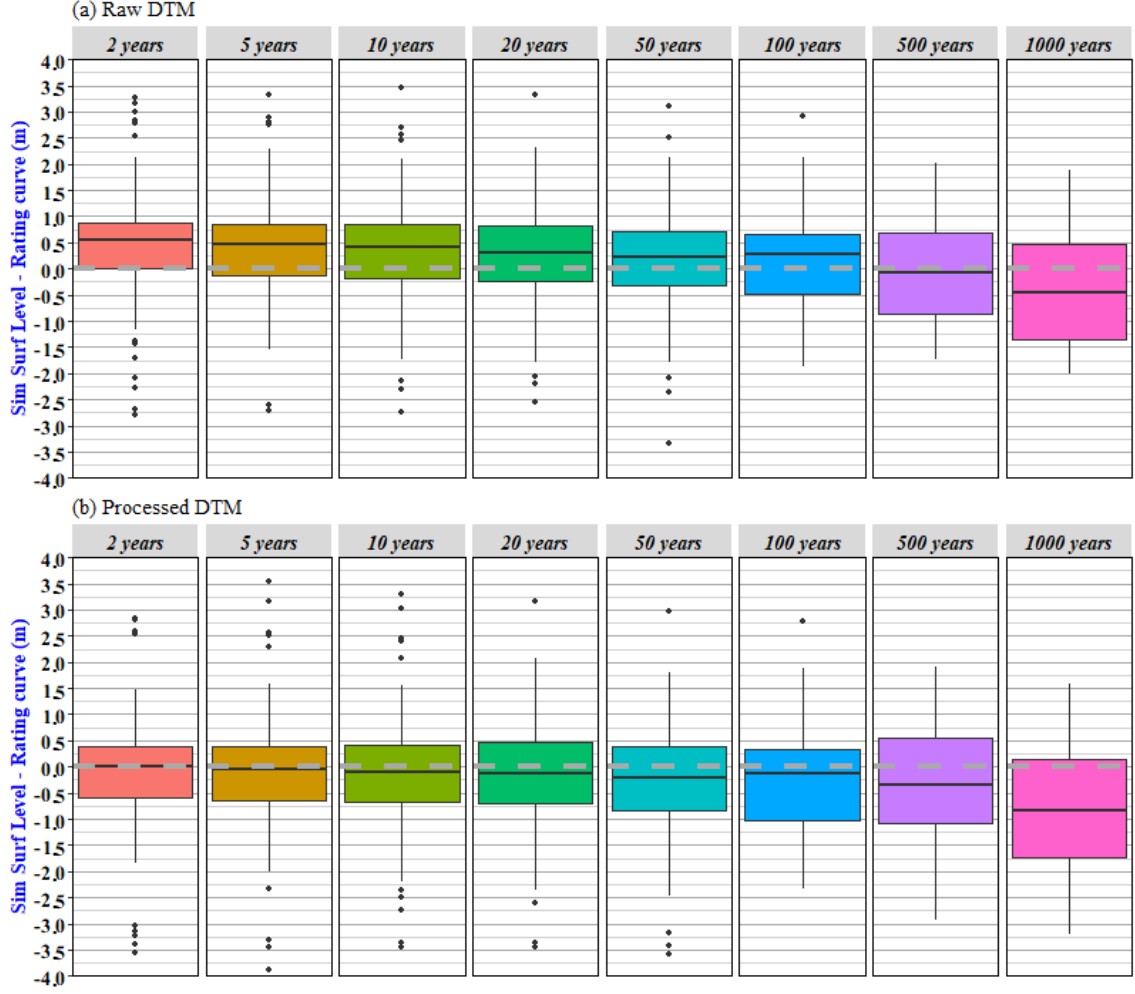

**Figure 8.** Boxplot of water altitude differences between the local expert rating curves and the regional 2D simulation results based on (a) the raw DTM and (b) the processed DTM for the eight considered return periods. The box plots represent, respectively, the 5 % and 95 % (whiskers) and the 15 % and 85 % quantiles (boxes).

The results show that the primary objective of the DTM preprocessing, i.e. reducing excessive overflow for low return periods, is reached. However, they also show that simulated water levels are less consistent with local theoretical rating curves for large return periods, i.e. 500-year and 1000-year. However, the boxplots for the 500-year and 1000-year return periods are only based on 33 and 20 values respectively (table 1), as few local forecasting services venture into this type of highly uncertain extrapolation of rating curves. As indicated in section 2.2.2, only 2% of directly measured (gauged) discharges exceed the 2-year return period, which does not allow for reliable extrapolations. Even when the rating curves suggest values for the 500-year and 1000-year return periods, this is systematically far outside of the reliability range defined by the local forecasting service. Finally, the spatial distribution of average errors has been explored. This does not reveal any clear spatial trend or concentration





of high errors values. This analysis did reveal that large differences (i.e. exceeding 2m) affect generally both the results obtained with the processed and raw DTMs. This indicates that the processing of the DTMs could not significantly reduce the distance between the locally established rating curve and the simulation results for the considered gauging stations, meaning that the source of the difference is probably not related to the limitations of the DTM.

## 4.2 Validation of flood event inundation maps using observed flood extents and high water marks

Readers are encouraged to refer to Hocini et al. (2021) for a more detailed explanation of the validation method used here, which consisted in comparing the inundation simulation results obtained for three specific events with observed flood extents and high water marks. In this section, we replicate this evaluation for our simulation results, obtained again with the Floodos hydraulic approach but using both an updated version of the raw DTM, and the processed version of this DTM. The results, shown in figure 9.b for the case of the Argens 2010 flood, align with those in figure 8, since they indicate a tendency to underestimate water levels during the most intense events (the three considered floods are major ones). This underestimation may be attributed to the uncalibrated Manning friction coefficients used or to the lack of representation of backwater effects from structures, among other factors.

Despite these limitations, the results showed a median (mean) altitude difference of -0.47 m (-0.57 m) for the Argens 2010 event, -0.30 m (-0.45 m) for the Aude 2018 event, and -0.31 m (-0.20 m) for the Alpes Maritimes 2015 event. These values fall within the typical error range of ± 0.3–0.5 m for validation based on this type of data, as noted by Bates (2022). The simulated inundation maps were also in good agreement with the observed flood areas (figures 9.a and 9.c), with CSI scores (measuring the overlap between observed and simulated flood extents) ranging from 55% to 85% for 90% of the river reaches studied. The median CSI values, close to 80%, are particularly satisfactory, considering that scores above 65% are generally regarded as indicators of reliable local estimates (Fleischmann et al., 2019).

## 5 Discussions

The flood hazard maps, obtained from automated computations implemented on large areas - which inevitably implies errors and uncertainties - resulted in very promising results when compared to the 171 available locally adjusted rating curves. The local comparison of simulated inundation maps with observed flood extents for three specific intense flood events also led to satifactory results, as already demonstrated by Hocini et al. (2021). However, these results also showed locally large differences between simulated and reference water altitudes (see table 2 for some examples in the case of flood hazard maps). This reveals some remaining limits which must be underlined.

A first explanation can be found in the DTM source. Notably, the entire study area was not covered by a Lidar DTM at the time of the calculations (Figure 10). Less accurate DTM production techniques, such as satellite radar and aerial photogrammetry, still predominate in some areas of the region under consideration. Table 2 indicates that among the 5% of stations with the largest errors, 6 out of 9 were located in areas without Lidar coverage. Figure 10 shows clear examples where the streambed is poorly defined or even non-existent in the original DTM. The pre-processing technique described in appendix C allows the



**Figure 9.** (a) Comparison of simulated and observed flood areas and water levels for the Argens 2010 event and the processed DTM. The dotted black rectangles refer to error clusters. (b) Comparison of simulated water levels based on the raw and processed DTMs, with observed high water marks (HWM) for the three events. (c) Comparison of simulated and observed flood extents for the Argens 2010 and the Aude 2018 events, using the CSI metric.

definition of a streambed when it is absent or poorly defined in the raw DTM. However it is not always able to compensate for all the limitations of an initially poorly defined DTM.

| Code | Name | Mean abs. difference (m) | % Lidar | % Photo | % Radar |
|------|------|--------------------------|---------|---------|---------|
| Y004501001 | Le Riu de Quérol à Porta | 16.04 | 0 | 100 | 0 |
| Y047406001 | La Têt à Saint-Féliu-d'Amont | 5.28 | 100 | 0 | 0 |
| V603501001 | Le Toulourenc à Malaucène | 5.14 | 65 | 0 | 35 |
| Y136401001 | Le Fresquel à Carcassonne | 5.10 | 100 | 0 | 0 |
| Y663501001 | La Bévéra à Sospel | 5.05 | 0 | 0 | 100 |
| Y040401001 | La Têt à Mont-Louis | 4.93 | 0 | 100 | 0 |
| V505401001 | L'Ardèche à Vallon-Pont-d'Arc | 3.95 | 0 | 100 | 0 |
| Y201002001 | L'Arre au Vigan | 3.71 | 0 | 100 | 0 |
| Y622401001 | La Tinée à Saint-Sauveur-sur-Tinée | 3.43 | 0 | 0 | 100 |

**Table 2.** The 5% of stations where the flood hazard maps showed the largest errors (calculated on the 2-y, 5-y, 10-y, 20-y return periods, with the results obtained from processed DTMs) and the source of the DTM slab they are located on.

As an illustration, the mean absolute errors calculated for the 2-year, 5-year, 10-year, and 20-year flood hazard maps across all stations is 1.03 meters, doubling to 1.95 meters for stations located in areas without Lidar coverage. This clearly demonstrates the advantages of Lidar technology. But even when Lidar measurements are available, achieving a perfect description of the geometry of streambeds on a large scale remains challenging, particularly due to the lack of bathymetric data, as the commonly used lasers do not penetrate the water column. Hocini et al. (2021) have also discussed main error sources affecting

their results on the simulation of historical events. They have observed that large errors are spatially clustered (see the rectangles in figure 9): they had already pointed out the limitations of the DTMs and particularly the lack of river bathymetry. The resulting underestimation of the streambed's cross-sectional area and hydraulic capacity will then depend on the season of survey and on the morphology of the river beds. Larger underestimations are more likely for winter surveys than for summer surveys. Similarly, ponds in the riverbed will be more frequent in the downstream sections of watercourses, often equipped

with engineered weirs or exhibiting a succession of pools and riffles. Ideally, Lidar surveys should be conducted during the Summer when water levels are low. However, the dense vegetation in Summer, which may partially or completely cover the streambed, especially in small rivers, can also pose problems for Lidar measurements.

Remaining errors can arise from a variety of other sources, including particularly geographical or altimetric referencing errors of sensors and/or HWM. In the case of station Y004501001 on the Riu de Quérol, which shows the largest mean

difference between the simulated water altitudes and the rating curve amongst all stations under study (average difference of -16m for the results obtained with the processed DTM), the exact location of the water level sensor can be questioned. The official altitude of the zero of the water level gauge is 1372m NGF on Hydroportail. According to the Hydroportail database, the streamgauge is located under a bridge which deck altitude is 1363m NGF. This clearly is inconsistent. The local slope of the stream is about 5%. The altitude of the zero level of the gauge, if reliable, thus suggests that the sensor should be located



several hundred meters upstream this bridge. Exchanges with the local hydrometric service could not confirm either of the two hypotheses : error in the reference altitude or location of the sensor. This example shows that despite a careful problem tracking with the help of local hydrometric services, some errors may remain.

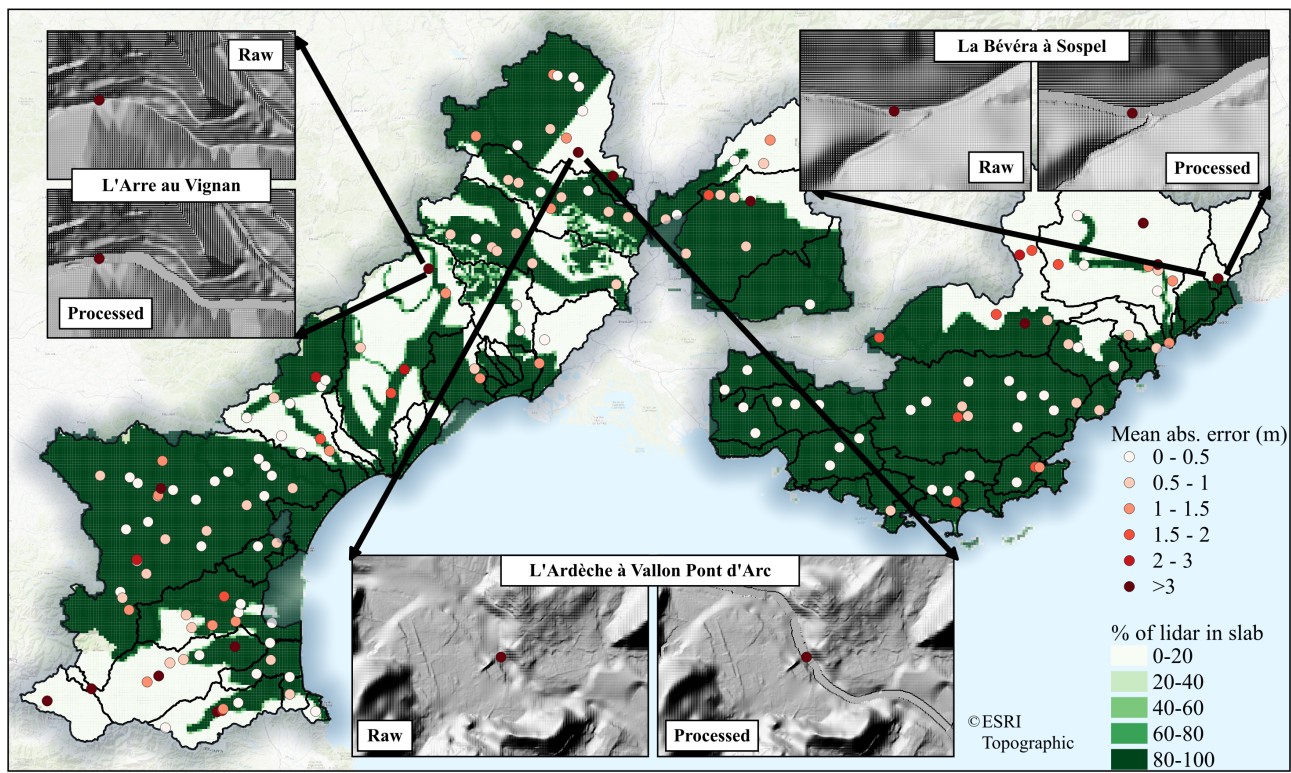

**Figure 10.** Portion of the area monitored by the radar technology, combined with the mean absolute errors between the simulated water altitudes and the rating curves (calculated on the 2-y, 5-y, 10-y, 20-y return periods), and examples where the streambeds where very poorly detected

It is worth noting that Figure 8 also shows that the chosen streambed/ flood plain couple of Manning's friction coefficients ($1/18$, $1/10$ $s.m^{-1/3}$) is adapted to low return periods, but that it leads to underestimate on average the water levels for larger re-

turn periods. If we trust the locally established rating curves, this suggests that for extreme flood events, the apparent friction of Mediterranean streambeds is certainly even greater than that adjusted in this application (rather $1/15$ than $1/18$ $s.m^{-1/3}$). This observation is consistent with the conclusions drawn by Lumbroso and Gaume (2012), who concluded, based on post-event survey data, that roughness coefficients should be much lower than the commonly recommended values (ranging from $1/30$ to $1/25$ $s.m^{-1/3}$), to provide realistic stage-discharge relations for extreme flash-flood events. Large regional implementations of

hydraulic models may also provide or confirm precious information about hydraulic properties of river beds.

# 6 Conclusions

This article documents a dataset including input and validation data, for the computation of flood hazard maps and inundation maps of historical floods over the French Mediterranean area. Reference computation results are also provided, using a the Floodos 2D hydraulic model, which solves the shallow water equations without inertia and in steady state regime. The article illustrates both the variety and richness of the proposed evaluation dataset in a region frequently affected by severe flash floods, and the relevance of approaches based on Lidar DTM and 2D numerical hydraulic models for high resolution regional flood mapping.

It was diagnosed that some major errors were linked to the DTM source, nevertheless other errors are probably introduced because of the simplicity of the chosen modelling approaches: simplified reservoir-based hydrological models, hydrodynamic method neglecting inertia terms and carried out independently for each reach, calculations in steady-state regime, monofrequency catalog of scenarios, etc. The proposed dataset provides the opportunity to address these issues and to assess the improvements of more complex approaches in the future. However, the results presented here, show that simple 2D models like Floodos, implemented on a regional scale, already deliver very satisfactory outcomes. These models are already effective for simulating historical flood extents and computing flood levels and hazard maps for reference discharge values. The challenging comparison with locally calibrated rating curves is particularly interesting, with an absolute mean difference of less than 0.5m between simulated levels and the reference rating curve in more than half of the cases, even for hazards of limited intensity (low return periods). This confirms the promising results already obtained by Hocini et al. (2021) based on the comparison with HWM levels for the three historical flash flood events.

*Data availability.* The dataset is provided on the French public platform Data gouv: https://doi.org/10.57745/IXXNAY (Nicolle et al., 2024).

*Author contributions.* The computational work was carried out by PN, NH, and JG. The original draft was written by JG, with contributions from EG, PN, and OP. PD, FP, PJ, PAG, and DL contributed original ideas and provided data. All authors participated in the reviewing and editing of the manuscript.

*Competing interests.* The contact author has declared that none of the authors has any competing interests.

*Acknowledgements.* The authors would like to thank the hydrometric services (SPC Grand Delta, SPC Méditerranée Ouest, DREAL PACA) for generously taking the time to discuss the rating curves. They also offer their warmest thanks to O. Delaigue, who helped extracting the gauging and rating curve data by providing R functions, and to F. Bourgin, who supported them in the use of the Cinecar rainfall-runoff model.



## Appendix A: Contents of the associated dataset

Table A1 provides a quick access guide to the dataset. Details on the attributes, units, sources, data size, formats, etc., can be
415   found in the readme.txt files provided with each data item, in the dataset.

| Data type | Data path | Signification | Section |
|---|---|---|---|
| Vector | 02_FLOOD_EVENTS / 01_INPUTS / DISCHARGE / RIVER_NETWORK | River network on which the CINECAR hydrological modelling was performed, for the simulation of the three historical flood events. | 2.3.1 |
| | 02_FLOOD_EVENTS / 03_VALIDATION / FLOOD_MARKS | High Water Marks collected for the three historical flood events. | 2.3.2 |
| | 02_FLOOD_EVENTS / 03_VALIDATION / OBSERVED_FLOOD_EXTENTS / ASSESSMENT_AREAS | Subcatchments affected by the Argens 2010 and Aude 2018 events, adapted to the observed flood extents. | 2.3.2 |
| | 02_FLOOD_EVENTS / 03_VALIDATION / OBSERVED_FLOOD_EXTENTS / OBS_FLOOD_SHP | Observed flood extents for the Argens 2010 and Aude 2018 events. | 2.3.2 |
| Raster | 01_FLOOD_HAZARD / 01_INPUTS / DTM / RAW | 5m DTMs resulting from a sampling of the 1m DTMs provided by IGN, for each area on which a flood hazard map is derived. | 2.2.1 |
| | 01_FLOOD_HAZARD / 01_INPUTS / DTM / PROCESSED | 5m DTMs that underwent the treatments described in section 3.1, for each area on which a flood hazard map is derived. | 2.2.1, 3.1 |
| | 02_FLOOD_EVENTS / 01_INPUTS / DTM / RAW | 5m DTMs resulting from a sampling of the 1m DTMs provided by IGN, for each catchment on which a historical flood event is simulated. | 2.3.1 |
| | 02_FLOOD_EVENTS / 01_INPUTS / DTM / PROCESSED | 5m DTMs that underwent the treatments described in section 3.1, for each catchment on which a historical flood event is simulated. | 2.3.1 3.1 |
| | 01_FLOOD_HAZARD / 01_INPUTS / HYDROGRAPHIC_NETWORK | Raster description of the minor bed for each area on which a flood hazard map is derived. | 2.2.1 |
| | 02_FLOOD_EVENTS / 01_INPUTS / HYDROGRAPHIC_NETWORK | Raster description of the minor bed for each catchment on which a historical flood event is simulated. | 2.3.1 |
| | 01_FLOOD_HAZARD / 01_INPUTS / DISCHARGE | Input SHYREG discharge quantiles for the simulation of flood hazard maps. | 2.2.1 |
| | 01_FLOOD_HAZARD / 02_OUTPUTS / WATER_LEVELS / RAW | Floodos flood hazard maps (water levels) obtained using raw DTMs. | 2.2.1 |
| | 01_FLOOD_HAZARD / 02_OUTPUTS / WATER_LEVELS / PROCESSED | Floodos flood hazard maps (water levels) obtained using processed DTMs. | 2.2.1 |
| | 02_FLOOD_EVENTS / 02_OUTPUTS / WATER_LEVELS / RAW | Floodos flood event maps (water levels) obtained using raw DTMs. | 2.3.1 |
| | 02_FLOOD_EVENTS / 02_OUTPUTS / WATER_LEVELS / PROCESSED | Floodos flood event maps (water levels) obtained using processed DTMs. | 2.3.1 |
| Table | 01_FLOOD_HAZARD / 03_VALIDATION / RATING_CURVES | Rating curve data for 171 gauging stations. | 2.2.2 |
| | 02_FLOOD_EVENTS / 01_INPUTS / DISCHARGE / HYDROGRAPH_TABLES / ... / hydrographs | Simulated discharge values at each 15 min time steps for the three historical events, on each river reach of the river network. | 2.3.1 |
| | 02_FLOOD_EVENTS / 01_INPUTS / DISCHARGE / HYDROGRAPH_TABLES / ... / lateral_hydrographs | Subcatchment lateral inflow values at each 15 min time steps for the three historical events, on each river reach of the river network. | 2.3.1 |

**Table A1.** Contents of the dataset



## Appendix B: Procedure applied on the rating curve data

The data provided in the dataset do not correspond to the raw data extracted from Hydroportail/BAREME - they have undergone a procedure to provide uniform (data extracted from Hydroportail are not formatted the same way as data extracted from BAREME), verified (some errors in the data could be detected), suitable (for flood hazard maps verification purposes), and simplified (keeping only the variable of interest) information. The procedure is as follows:

- Concerning the locations: stations outside the study area (Figure 2) were removed. Stations that could not be connected to a pixel of the stream network (distance criterion = 100m) were also removed. Stations that could not be connected to a SHYREG pixel (distance criterion = 150m) were removed. Remaining stations were manually relocated, and the positions were checked with the local hydrometric services. This manual verification can be time-consuming but is crucial, especially for stations located near confluences, which can be attributed to the wrong river reaches. It is important to note that the coordinates of a station do not always correspond to the location of the measurement device, but sometimes to the location of the recording or remote connecting box, that can be located several hectometres from the surveyed river cross-section.

- Concerning the values: stations without a zero reference altitude of the water level gauge were removed, and remaining values were checked: obvious altimetric errors were corrected, and less obvious but detectable errors were discussed with the local hydrometric services. Stations without any recent rating curve (date threshold = 2010) were removed. Rating curves without any point higher than the 2-year return period were removed. In case of several available rating curves for the same gauging station, only the most recent curve was considered.

## Appendix C: Brief description of the DTM preprocessing method

The DTM preprocessing procedures carried out in this study are very similar to those described by Wimmer et al. (2021), they are organised here into four main stages:

1. Correcting the river axes. To achieve this, orthogonal cross sections were extracted every 1m on the original river network (extracted from BD TOPAGE®, see section 2.2.1), resulting in a cross-sectional profile function $h(d)$ where $d$ is the signed distance orthogonal to the river axis. Then, six weight functions accounting for different geometric and hydraulic criteria (such as the cross section form, water flow properties, etc.) are defined and combined into a global weight function $w_{final}$. If there exists a $d_{max} > 0$ verifying $w_{final}(d_{max}) = max(w_{final})$, then this $d_{max}$ defines the new location of the corrected river axis, for the considered cross section. These treatments only concern the hydrographic network, and thus keep the DTMs unchanged.

2. Detecting the riverbanks. In order to do this, we work on the same cross sections as before, and we translate the expected shape of the upper edge of embankment (strong negative curvature due to the flattening above the riverbanks) into a

mathematical requirement (high first and small second derivative of the cross-sectional profile). Compared to Wimmer et al. (2021), an additional criterion (threshold on the break of slope) was used to better represent V-shaped small rivers.

3. Validating the riverbanks detection. While the detection of riverbanks resulted in satisfying delineations of the streambeds for a high proportion of cross sections, some erratic behaviours have been observed, especially in case of highly vegetated riverbanks. Several criteria were used to decide whether to eliminate a cross section: intersection with water bodies, several intersections with another river, intersection with bridges, buildings, sports grounds, roads, railways, inconsistency with reference river width classes, detected river width as large as the initial cross section.

4. Reshaping the streambed. First, the positions of the right and left banks of the streambed were smoothed with a moving average procedure implemented on the cross-sectional coordinates of the 5 closest upstream and downstream cross-sections. Then, the longitudinal profile of the minimal altitudes of all cross-sections was checked and local maximums were replaced by linearly interpolated values to produce a river bed with negative longitudinal profile slopes. Finally, in the DTMs, the minimal altitudes of the each cross sections were applied to the whole width of the detected proportion of the stream bed located between the river banks.

Figure C1 illustrates the results on a 2 km-long reach of the Orb river, upstream of the city of Béziers. In this case, the errors in the raw DTMs, identifiable by the triangular shapes in the streambed, were probably caused by interpolation errors (presumably in the absence of lidar points).

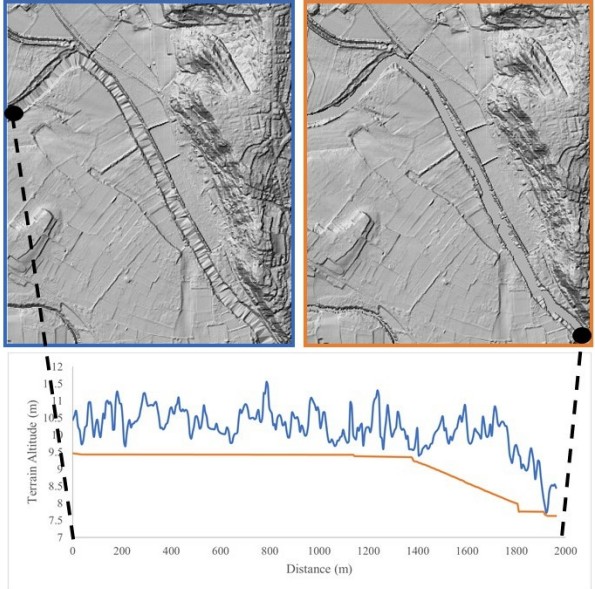

**Figure C1.** Illustration of the impact of the DTM processing on the Orb river (near the city of Béziers) streambed cross-sectional and longitudinal profiles: original (left) and processed (right) DTMs and corresponding original (blue) and processed (orange) longitudinal profiles.





**Appendix D: Details on the implementation of Floodos in this study**

Constraints linked to the amount of data to be handled first led to divide the domain into 61 subdomains (see figure 2), and then to divide the river network of each subdomain into river reaches. This division was initially made at the confluences, and
river reaches longer than 10km were divided again, the objective being to create calculation domains with a number of cells not exceeding $25 \times 10^6$. In the end, the average length of a river reach is about 5km. Each river reach was extended by 500m downstream, in order to limit the influence of the dowstream boundary condition on the simulation results. The extent of the domain was also extended by 2km on each side, in order to reduce inconsistencies at confluences (in particular due to possible backwater influences), and to ensure that the computation domain limit is located sufficiently far downstream of the considered
river reach. The mesh underwent other treatments, such as the addition of artificial borders upstream the computation domains to force the flow downstream the river reach, and the change (decrease) of elevation values for cells in the Mediterranean sea, to prevent uncontrolled backwater effects at the coastline . These treatments led to the creation of the topography ".alt" file.

The discharges ".rain" file is a grid which values define the discharges to be injected at each injection point. These injection points, corresponding to 5m resolution pixels, need to be correctly located (i.e., on the correct minor bed, which can be tricky
near confluences), even though the input SHYREG or CINECAR discharges are originally given at the 50m resolution. Several methods are possible to address this issue, see (Godet et al., 2024). In this work, points 50m apart were drawn along the hydrographic network, and SHYREG discharges were extracted at these points. The first discharge injection is made on the most upstream point of the river reach, and corresponds to the total discharge circulating in the river reach. Downstream of that point, all injections are increments.

The main challenge concerns the adjustment of the parameters controlingthe convergence of computations. The elementary volume corresponding to each precipiton should be carefully fixed to ensure convergence and computation speed. Davy et al. (2017) have defined a maximum possible value for the volume of the precipiton $V^p$ ensuring the convergence:

$$V^p = 0.75 \times S_0 \times \Delta x^3 \tag{D1}$$

Where $S_0$ is the water surface slope. Several tests aiming at finding a compromise between fast calculations and correct
convergence led to the following solution, adapted from (Hocini, 2022): during the initialisation phase of each simulation, the precipiton volume will be progressively reduced to $0.0625\text{m}^3$, which respects the condition of equation D1 as long as $S_0$ is lower than 0.07%. The duration of the initialisation phase, as well as the total duration of the simulation are indicated in table D1, and are defined in Calculation Time Units (CTU). 1 CTU refers to the injection of a fixed number of precipitons. The convergence has been verified on the last 25 CTU of the simulation. The water level was considered as stabilised if the
variation was lower than 1mm between iterations.

Calculations were parallelised on a 20 cores and 128 GB RAM cluster and took roughly two months.



| Parameter | Signification | Value |
|---|---|---|
| $CTU_{coeff}$ | Average number of precipitons brought during 1 CTU on one injection point | 40 |
| $step$ | Number of steps in the initialisation phase | 4 |
| $N_{CTU,init}$ | Length, in CTU, of the initialisation phase | 10 |
| $V_{start}^{P}$ | Initial volume of precipiton in m3 | 0.25 |
| $V_{end}^{P}$ | Final volume of precipiton in m3 | 0.0625 |
| $N_{CTU}$ | Number of CTU for each simulation | 100 |
| $Cond$ | Max. authorized error for difference between water level and mean water level during the last 25 CTU, in meters | 0.001 |

**Table D1.** Tables of parameters related to the convergence of the model

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
