# Peer review of "Benchmark dataset for hydraulic simulations of flash floods in the French Mediterranean region"

_Earth System Science Data, 2024_

## Author Comment (AC1)

**Authors' Response to Reviews of**

**Benchmark dataset for hydraulic simulations of flash floods in the French Mediterranean region**

Juliette Godet, Pierre Nicolle, Nabil Hocini, Eric Gaume, Philippe Davy, Frederic Pons, Pierre Javelle, Pierre-André Garambois, Dimitri Lague, and Olivier Payrastre
*Earth System Science Data*, `https://doi.org/10.5194/essd-2024-472`
* * *
**RC:** *Reviewers' Comment*,  AR: Authors' Response,  ☐ Manuscript Text

**RC:** *This manuscript describes a comprehensive benchmark dataset related to the study of flash flood events in the Mediterranean region of France. Specifically, the dataset comprises the input and output data used for the hydraulic simulations of three events, as well as a range of data for model validation. I believe that the dataset that might be of interest of other research groups working on flash flood hazard and risk in the Mediterranean Region, and therefore I recommend its publication, after having addressed a few minor remarks.*

AR: Dear Francesco Dottori,

Thank you for accepting to review our article and for the useful comments you have provided. We will explain hereafter how we plan to adapt the manuscript according to your recommendations.

**RC:** *Page 6 L7: can you please provide a reference or a link for the BD TOPAGE hydrographic network database?*

AR: It was provided in hyperlink, but it has been made more visible:

> ..., derived from BD TOPAGE®, which is the French reference hydrographic network database, accessible at `https://www.data.gouv.fr/fr/datasets/bd-topage-r/` (last accessed 14/03/2025).

**RC:** *Page 7 L26-36: this paragraphs refers to parts of the dataset that are only describer later on in the manuscript. you might want to move it to Section 3.3, or to incorporate the information in Figure 1*

AR: In this paragraph, we aim to present the output data from the flood hazard simulation. As shown in Figure 1, both inputs and outputs are described in Section 2.2.1. Furthermore, we believe this paragraph is important for introducing the DTM preprocessing method we applied. Therefore, while we appreciate your suggestion, we propose to retain this paragraph in its current position.

**RC:** *Page 16, L322-324: other possible reasons for the observed differences between rating curves and simulations could be: the use of steady state simulations / the approximation given by the inertia-only version of SWE, which might not reproduce well water depths where there are large changes in flow section*

AR: These limitations are mentioned in the conclusions section (Line 395) and in the section explaining the Floodos computations (Section 3.3). However, we agree that they can also be briefly recalled here:

> ... is probably not related to the limitations of the DTM. As highlighted in Section 3.3, other sources of error may stem from neglecting inertia terms in the resolution and assuming a steady-state regime.

**RC:** *Page 7 L45:"The rating curves are also subject to uncertainty, as they are derived from measured discharge*

*data''. You might want to refer her to the work by Di Baldassarre and Montanari (2009), who provided a quantification of the overall uncertainty of discharge estimates from rating curves*

AR: Thank you for your suggestion. We will incorporate this reference into the text, along with other relevant studies on the subject:

> The rating curves are also subject to uncertainty, as they are derived from measured discharge data, which themselves carry significant uncertainty (Di Baldassarre and Montanari, 2009). Notably, tools exist to quantify the uncertainty associated with hydrometric rating curves (Le Coz et al., 2014). In this study, we argue that rating curves provide a convenient and independent data source for evaluation, despite their inherent uncertainties, which should be considered when analyzing the results.

RC: *Section 4.2: Could you please elaborate on the potential influence of solid transport and related erosion/deposition processes on flood extent and water depths? Do you think that these processes could have played a role in the three case studies?*

AR: In these three events, solid transport was limited. However, localized blockages were observed, as reported by Hocini et al. (2021). We will incorporate this into the text as follows:

> Although the influence of solid transport related to erosion and deposition processes was limited during these three events—unlike a more recent event, the Alex storm in October 2020, which severely affected the study area (see Payrastre et al., 2022)—localized blockages were observed during the Aude 2018 event, as documented by Hocini et al. (2021). These blockages may have contributed to significant underestimations of water levels and flood extent in the vicinity of the blocked bridges.

RC: *Page 19 L383 "...that roughness coefficients should be much lower than the commonly recommended values.." perhaps did you mean higher? (Manning's coefficient increases with increasing roughness)*

AR: Yes, thank you for catching this. We were considering the Strickler coefficient, which led to the confusion. We have corrected this in the text.

RC: *Page 20 L388, typo: using the Floodos 2D hydraulic model*

AR: Thank you.